# Experimental Conditions That Influence the Utility of 2′7′-Dichlorodihydrofluorescein Diacetate (DCFH_2_-DA) as a Fluorogenic Biosensor for Mitochondrial Redox Status

**DOI:** 10.3390/antiox11081424

**Published:** 2022-07-22

**Authors:** Lianne R. de Haan, Megan J. Reiniers, Laurens F. Reeskamp, Ali Belkouz, Lei Ao, Shuqun Cheng, Baoyue Ding, Rowan F. van Golen, Michal Heger

**Affiliations:** 1Jiaxing Key Laboratory for Photonanomedicine and Experimental Therapeutics, Department of Pharmaceutics, College of Medicine, Jiaxing University, Jiaxing 314001, China; l.dehaan@erasmusmc.nl (L.R.d.H.); m.reiniers@haaglandenmc.nl (M.J.R.); aolei@zjxu.edu.cn (L.A.); lena_310@zjxu.edu.cn (B.D.); 2Laboratory for Experimental Oncology, Department of Pathology, Erasmus MC, 3015 GD Rotterdam, The Netherlands; 3Department of Surgery, Haaglanden Medisch Centrum, 2262 BA The Hague, The Netherlands; 4Membrane Biochemistry and Biophysics, Department of Chemistry, Faculty of Science, Utrecht University, 3584 CH Utrecht, The Netherlands; 5Department of Vascular Medicine, Amsterdam University Medical Centers, Location AMC, 1105 AZ Amsterdam, The Netherlands; l.f.reeskamp@amsterdamumc.nl; 6Department of Medical Oncology, Amsterdam University Medical Centers, Location AMC, University of Amsterdam, Cancer Center Amsterdam, 1105 AZ Amsterdam, The Netherlands; a.belkouz@amsterdamumc.nl; 7Department of Hepatic Surgery VI, The Eastern Hepatobiliary Surgery Hospital, The Second Military Medical University, Shanghai 200438, China; chengshuqun@smmu.edu.cn; 8Department of Gastroenterology and Hepatology, Leiden University Medical Center, 2333 ZA Leiden, The Netherlands; r.f.van_golen@lumc.nl; 9Department of Pharmaceutics, Utrecht Institute for Pharmaceutical Sciences, Utrecht University, 3584 CG Utrecht, The Netherlands

**Keywords:** fluorogenic redox probe, fluorescence imaging, rotenone, antimycin A, myxothiazol, piericidin A, electron transport chain inhibitors, oxidative stress, hepatocytes

## Abstract

Oxidative stress has been causally linked to various diseases. Electron transport chain (ETC) inhibitors such as rotenone and antimycin A are frequently used in model systems to study oxidative stress. Oxidative stress that is provoked by ETC inhibitors can be visualized using the fluorogenic probe 2′,7′-dichlorodihydrofluorescein-diacetate (DCFH_2_-DA). Non-fluorescent DCFH_2_-DA crosses the plasma membrane, is deacetylated to 2′,7′-dichlorodihydrofluorescein (DCFH_2_) by esterases, and is oxidized to its fluorescent form 2′,7′-dichlorofluorescein (DCF) by intracellular ROS. DCF fluorescence can, therefore, be used as a semi-quantitative measure of general oxidative stress. However, the use of DCFH_2_-DA is complicated by various protocol-related factors that mediate DCFH_2_-to-DCF conversion independently of the degree of oxidative stress. This study therefore analyzed the influence of ancillary factors on DCF formation in the context of ETC inhibitors. It was found that ETC inhibitors trigger DCF formation in cell-free experiments when they are co-dissolved with DCFH_2_-DA. Moreover, the extent of DCF formation depended on the type of culture medium that was used, the pH of the assay system, the presence of fetal calf serum, and the final DCFH_2_-DA solvent concentration. Conclusively, experiments with DCFH_2_-DA should not discount the influence of protocol-related factors such as medium and mitochondrial inhibitors (and possibly other compounds) on the DCFH_2_-DA-DCF reaction and proper controls should always be built into the assay protocol.

## 1. Introduction

Reactive oxygen species (ROS) are radical and pro-oxidant derivatives of oxygen that are important in biological systems [1,2]. Under physiological conditions, ROS regulate cellular redox homeostasis [2,3] and play a critical role in signal transduction [4,5]. Under pathological conditions, the redox homeostasis becomes perturbed when ROS are produced in excess and/or when the antioxidant machinery is compromised. The imbalance between ROS production and detoxification is known as oxidative stress and can lead to irreversible damage to DNA, proteins, and lipids [6,7,8]. Oxidative stress has been causally related to various neurodegenerative [9,10,11], metabolic [7,12], inflammatory [8,13,14], malignant [15,16,17], and atopic disorders [18] and hence constitutes an important area in basic and translational research.

The chief intracellular site of ROS production is the mitochondrion, where a small fraction of consumed oxygen is continuously reduced to superoxide (O_2_^•–^) by electrons leaking from the electron transport chain (ETC) [19,20]. The extent of O_2_^•–^ formation is exacerbated under certain pathological conditions, causing O_2_^•–^ and derivative ROS to reach toxic levels and induce (extensive) mitochondrial and cellular damage. A frequently employed method to study mitochondrial oxidative stress is to simulate the in vivo situation in cell systems using ETC inhibitors. The inhibition of selected sites in the ETC with, for example, rotenone (complex I) or antimycin A (complex III) interrupts electron flow and induces leakage of electrons from the inhibited complex, thereby mimicking the mitochondrial formation of ROS under the aforementioned pathological conditions.

Analytical methods to determine mitochondrial ROS production in vitro are essential in studying oxidative stress-related diseases. A large number of fluorogenic probes have become available for these purposes. The probes are mostly non-fluorescent small molecules that are oxidized to stable fluorophores by ROS [21,22]. As these biosensors are generally non-toxic, easy to use, relatively cheap, and can be measured in real-time using basic laboratory equipment, the application of fluorogenic probes for assessing oxidative stress has considerably increased in the past decade [21,23,24]. In that respect, one of the most widely used probes is 2′,7′-dichlorodihydrofluorescein diacetate (DCFH_2_-DA). DCFH_2_-DA diffuses into cells where it is deacetylated by esterases to the non-fluorescent 2′7′-dichlorodihydrofluorescein (DCFH_2_) (Appendix A). The deacetylation causes the probe to be retained in the cytosol, where it is subsequently oxidized to the fluorescent 2′7′-dichlorofluorescein (DCF) by ROS [25,26,27] that is produced by, for example, the electron transport chain (Appendix A). DCFH_2_ reacts with multiple types of ROS [25,28,29,30], albeit at different rate constants, and hence acts as a generic ROS probe [28,31]. Accordingly, the magnitude of DCF fluorescence is directly proportional to the extent of intracellular oxidative stress [32].

As was previously emphasized by the editors of Free Radical Biology and Medicine [33], the interpretation of data that were generated with fluorogenic probes such as DCFH_2_-DA is complex and the protocol should always follow strict rules to prevent misinterpretation of experimental results [21,22,34]. Accordingly, several factors must be taken into account that can influence the formation and/or the fluorescence of the probe’s end product. For example, the effects of the assay system (e.g., medium, compounds), the loading temperature, and the pH on the formation and fluorescence of DCF under cell-free conditions should be determined to ensure that the readout is valid [28,31,35,36]. Factors such as these may seem trivial and are too frequently disregarded in published studies in which fluorogenic redox biosensors were used. While studying immune signaling by oxidatively stressed cells, the practical implications that were postulated by Forman et al. [33] became overly clear when we observed that the rate and extent of ROS production in cells that had been subjected to medium containing DCFH_2_-DA and rotenone was equivalent to the rate and extent of ROS production in medium containing DCFH_2_-DA and rotenone but in the absence of cells, ultimately invalidating the results that initially supported our hypothesis. 

Experiments were therefore conducted to demonstrate that (1) various ETC inhibitors mediate the DCFH_2_-DA → DCF conversion in a dose-dependent manner in the absence of cells and (2) the extent of DCF formation depends on the type of culture medium that is used, the pH of the assay system, the presence of fetal calf serum (FCS), and the final DCFH_2_-DA solvent concentration. In light of these findings, the DCFH_2_-DA assay was optimized under cell-free conditions. Next, the ability of ETC inhibitors to induce oxidative stress was investigated in three hepatocyte cell lines using the optimized protocol. Hepatocyte cell lines were deliberately chosen inasmuch as hepatocytes are replete with mitochondria [37] and because liver cell lines are commonly used for various research purposes. The most important conclusion of the study is that experiments with DCFH_2_-DA should not discount the influence of ancillary factors such as medium and mitochondrial inhibitors (and possibly other compounds) on the DCFH_2_-DA → DCF reaction and that proper controls should always be built into the assay protocol. 

## 2. Materials and Methods

Online supplemental figures and tables are designated with prefix ‘S’. References used in the Appendix A have been embedded in the bibliography of the main text [38,39,40,41,42,43,44,45,46,47,48,49,50,51,52,53,54,55,56,57,58,59,60,61,62,63,64].

### 2.1. Chemicals and Reagents

Chemicals and reagents are listed in Appendix A. All of the experiments were performed in flat bottom 24-wells plates (Corning, Corning, NY, USA) or in flat bottom 96-wells plates (Greiner Bio-One, Kremsmuenster, Austria). Dilutions were made in 50-mL polypropylene tubes and 12-mL polystyrene tubes, both from Greiner Bio-One. ETC inhibitors were dissolved in DMSO (rotenone, myxothiazol, and piericidin A) or ethanol (antimycin A) to concentrations of 2.0 mM (rotenone), 2.5 mM (myxothiazol), 1.7 mM (piericidin A), and 45.6 mM (antimycin A). As ETC inhibitors in general and piericidin A in particular are extremely toxic to humans and animals, the chemicals were handled and stored with special precautions in accordance with institutional policies. 

All the concentrations throughout the manuscript refer to the final concentration unless otherwise indicated.

### 2.2. Cell Culture

The murine non-transformed hepatocyte cell line AML-12 was provided by Riekelt Houtkooper (Amsterdam University Medical Centers, location AMC). The human hepatoma cell line HepaRG was provided under end-user license to Ruurdtje Hoekstra by Guguen Guillouzo (INSERM, Rennes, France). HepG2 human hepatoma cells were obtained from American Type Culture Collection (Manassas, VA, USA). The cells were grown under standard culture conditions (37 °C, humidified atmosphere composed of 5% CO_2_ and 95% air) in phenol-red containing supplemented William’s E (WE) medium as described [31]. The cells were grown in 75 cm^2^ culture flasks (Corning, Corning, NY, USA) and received fresh culture medium twice a week. After 2-4 weeks (HepaRG) or 5 days (AML-12, HepG2) of culture, the cells were washed twice with phosphate buffered saline (PBS) and detached in a mixture of Accutase, Accumax, and PBS (2:1:1 volume ratio) for 10–15 min at 37 °C. HepaRG cells were subcultured at a split ratio of 1:5–1:6, seeded in 24-wells plates, and used for experiments 28–32 days after seeding to allow for optimal hepatocyte differentiation [31,65]. AML-12 and HepG2 cells were also subcultured at a split ratio of 1:5–1:6 and seeded in 24-wells plates. In contrast to HepaRG cells, HepG2 and AML-12 cells were used for experiments immediately after reaching 100% confluence.

### 2.3. Kinetics Measurement of DCF Fluorescence (Cell-Free)

All kinetics experiments were performed in 24-wells plates using an assay volume of 500 μL per well, containing either 0.5 μL of 50 mM DCFH_2_-DA in DMSO (50 μM final probe concentration) or 0.5 μL of DMSO (solvent control). The DCF fluorescence was recorded using a temperature-controlled (37 °C) Synergy HT microplate reader (BioTek Instruments, Winooski, VT, USA) set to an excitation wavelength (λ_ex_) of 460 ± 40 nm and an emission wavelength (λ_em_) of 520 ± 20 nm. DCF fluorescence was measured in kinetics mode for 2 h at 15-min intervals using the bottom voxel read setting. All fluorescence measurements were performed in non-supplemented and phenol red-lacking WE medium containing 25 mM HEPES (assay medium) unless otherwise indicated. 

The influence of ETC inhibitors on DCF formation was measured over a concentration range of 0–30 μM (antimycin A), 0–100 μM (rotenone), 0–10 μM (myxothiazol), and 0–10 μM (piericidin A) at three pH values (pH = 6, pH = 7.4, and pH = 9) and in unbuffered assay medium. The inhibitor concentrations and incubation times were selected to fall within the range of published protocols (Appendix A, [38,39,40,41,43,44,45] (rotenone), [46,47,48,49,50,51,52,53] (antimycin A), [48,54,55,56,57,58] (myxothiazol), and [59,60,61,62] (piericidin A)). The final solvent concentration was held constant at 0.1% (*v*/*v*) DMSO or 0.2% (*v*/*v*) ethanol for all ETC inhibitor concentrations to ensure no deleterious effects on cell viability (Appendix A). 

The influence of different culture media (WE, DMEM, RPMI 1640, and DMEM/Ham’s F12, all containing 25 mM HEPES buffer) and solvent concentrations (DMSO, ethanol, and methanol, all 0–10% (*v*/*v*) in assay medium) on DCF formation was tested at pH = 7.4. The effects of FCS, heat-inactivated FCS (30 min at 56 °C, both 0–10% *v*/*v*), and bovine serum albumin (BSA; 0–3.4 g/L) on DCF formation were assessed in 25 mM HEPES buffer (pH = 7.4). The BSA concentration range was selected on the basis of the albumin concentration of the FCS lot (34 g/L) as specified in the manufacturer’s certificate of analysis. The influence of pH on DCF formation was tested in 25 mM HEPES buffer that was titrated to pH = 6, pH = 7.4, or pH = 9. The impact of buffers on the DCFH_2_-DA assay was tested at TRIS and HEPES concentrations of 5, 10, and 25 mM (pH = 7.4).

### 2.4. Kinetics Measurement of DCF Fluorescence in Hepatocytes after Physiological Buffer Exposure

Hepatocytes were grown to confluence in 24-well plates as described in Section 2.2. The cells received 500 μL of assay medium, PBS, or Hank’s balanced salt solution (HBSS, Lonza) that was supplemented with 50 μM DCFH_2_-DA (from a 50 mM stock in DMSO), or assay medium (pH = 7.4) containing 0.1% (*v*/*v*) DMSO (n = 6/group) as a control. DCF fluorescence was subsequently recorded for 2 h on a Synergy HT microplate reader (BioTek Instruments) as described in Section 2.3, after which the cells were washed with PBS and incubated with 310 μL of assay medium (pH = 7.4) containing 10% (*v*/*v*) WST-1 reagent (Roche Applied Science, Penzberg, Germany) for 15 min under standard culture conditions. After the incubation period, 210 μL of medium was transferred to a 96-wells plate and absorption was immediately read at 450 nm on a microplate reader (BioTek Instruments) as a measure for mitochondrial activity. The DCF fluorescence and WST-1 absorption were corrected for the DNA content per well as described in Section 2.6. The WST-1 results were normalized to the results of the control (i.e., medium) group. 

### 2.5. Kinetics Measurement of DCF Fluorescence in Hepatocytes after ETC Inhibition

AML-12, HepG2, and HepaRG cells were grown to 100% confluence as described in Section 2.2, after which the culture medium was replaced with fully supplemented WE medium containing ETC inhibitors as described in Section 2.3. 

The cells were incubated with ETC inhibitors for 0–48 h under standard culture conditions. At the start of the experiment, the cells were washed twice with PBS equilibrated at 37 °C after the incubation period. For the 0-h incubation experiments, the cells received 500 μL of unbuffered assay medium containing 0.5 μL of 50 mM DCFH_2_-DA in DMSO (50 μM final probe concentration) and 0.5 μL of rotenone in DMSO (0–100 μM final rotenone concentration) or 1 μL DMSO (solvent control) and were immediately placed in the plate reader. For longer incubation experiments, the cells received 500 μL of assay medium containing 0.5 μL of 50 mM DCFH_2_-DA in DMSO or 0.5 μL of DMSO (solvent control). The fluorescence for all incubation times was subsequently measured for 2 h at λ_ex_ = 460 ± 40 nm and λ_em_ = 520 ± 20 nm. Immediately after the read, the experiment was duplicated in a cell-free 24-well plate that was loaded with the aforementioned probes at equimolar concentration in 500 μL of assay medium (n = 4) or 0.1% (*v*/*v*) DMSO in 500 μL assay medium (solvent control, n = 4). For the 0-h incubation experiment, the cell-free plate was loaded with 50 μM DCFH_2_-DA and 0–100 μM of rotenone in 500 μL assay medium (n = 4 per rotenone concentration) or 50 μM DCFH_2_-DA and 0.2% (*v*/*v*) DMSO in 500 μL assay medium (solvent control, n = 4). For each time point, the fluorescence emission from the cell-free plate was subtracted from the fluorescence that was obtained in the plate with cells to correct for DCFH_2_-DA autoxidation and background fluorescence.

### 2.6. DNA Quantification

To normalize the DCF fluorescence to the number of cells per well, the DNA content of each well was determined using the Hoechst assay. The cells were washed twice with PBS immediately after the kinetics read and lysed for ≥1 h in 0.2 M NaOH at 37 °C. The lysate was resuspended by pipetting and 10 μL of lysate was transferred in duplicate to a 96-well plate. Subsequently, 200 μL of freshly prepared working reagent (1:1 mixture of 4 M NaCl and 0.1 M PO_4_ buffer (pH = 7.4) that was supplemented with 0.1 μg/mL Hoechst 33342 (from a 200 μg/mL stock) was added to each well, after which fluorescence was read at λ_ex_ = 340 ± 30 nm and λ_em_ = 460 ± 40 nm on a Synergy HT microplate reader (BioTek Instruments). The DNA concentration per well was calculated on the basis of a standard curve that was comprised of known concentrations of herring sperm DNA (0–250 ng/μL DNA in 0.2 M NaOH). 

### 2.7. Statistical Analysis

Statistical analyses were performed using GraphPad Prism (GraphPad Software, La Jolla, CA, USA). The data were tested for intragroup differences using a one-way ANOVA with Dunnett’s (when all groups were compared to a control group) or Tukey’s (when all groups were compared to each other) post hoc test. Additional details on the statistical analysis are provided in the figure legends. A *p*-value of ≤ 0.05 was considered statistically significant.

## 3. Results and Discussion

DCFH_2_-DA is a fluorogenic probe that is used to detect intracellular ROS formation in vitro and in vivo [28,66]. The change in fluorescence intensity (∆flu) of DCF is proportional to the extent of DCFH_2_ oxidation by ROS but also by other compounds (e.g., ETC inhibitors and constituents in the cell culture medium, as demonstrated below). 

Kinetics measurements in preliminary cell-based experiments revealed that differences in ∆flu (the difference in DCF fluorescence between t = 0 and t = 120 min) become manifest after ~30 min. To obtain statistically significant results, longer read times (i.e., 2 h) were required. However, during a 2-h read, the cells are subjected to experimental conditions that are not necessarily compatible with their preferred/native environment, given that most commercial plate readers are not equipped to provide standard culture conditions during a kinetics read. DCF fluorescence may, therefore, reflect phenomena other than those that are attributable to the compound or process under investigation. 

Building on previous methodological work [28,31,67,68,69], we first elaborated on the influence of mitochondrial agents on the DCFH_2_-DA → DCF conversion in medium, which is relevant for in vitro mitochondrial research. Based on the findings, an assay system was designed that minimizes undesired influences on DCF formation in cells. The assay system was subsequently used to investigate the effects of ETC inhibitors on mitochondrial ROS formation in cultured hepatocytes. Specifically, the experiments were conducted to demonstrate the effect of various protocol-related factors on the DCFH_2_-DA → DCF reaction. Readers should note that the purpose of these experiments was not to provide a mechanistic underpinning of (inexplicable) data. This approach was deliberate as the value of such mechanistic insight is limited and would distract readers from the central message.

### 3.1. Rotenone Dose-Dependently Induces DCFH_2_-DA → DCF Conversion in Cell Culture Medium

The semiquinone antagonist rotenone is one of the most frequently used complex I inhibitors. Rotenone diffuses across the cell membrane and the mitochondrial outer membrane and binds to complex I, deterring the flow of NADH-derived electrons through the complex [67] (Appendix A). As a result, these electrons reduce molecular oxygen to O_2_^•–^ and induce mitochondrial oxidative stress when the O_2_^•–^ or its derivative hydrogen peroxide (H_2_O_2_) are not sufficiently dismutated. This scenario is analogous to several oxidative stress-related conditions that culminate in disease [12]. The conditions include hepatic cholestasis, where hydrophobic bile salts intercalate in the inner mitochondrial membrane of hepatocytes [70] and induce ETC uncoupling [71] and mitochondrial oxidative stress as a result of excessive electron leakage [72], as well as hepatic- [73,74], cardiac- [75,76,77], renal- [78], and brain-ischemia/reperfusion injury [79]. The priming of cells with rotenone is therefore a suitable model to emulate conditions of mitochondrial oxidative stress [54] and to study downstream biological and biochemical processes. DCFH_2_-DA can be employed to correlate the extent of oxidative stress to the ramifications of mitochondrial oxidative stress, such as the mode of cell death or inflammatory signaling [14]. 

In this framework, AML-12 hepatocytes were incubated simultaneously with rotenone and DCFH_2_-DA in accordance with the literature [31,67] to determine the extent of ROS formation as a function of time. Initially, the hepatocytes appeared to produce ROS as a by-product of normophysiological function (0 µM rotenone, Figure 1A) [19], which seemed to be exacerbated at a rotenone concentration of ≥10 µM in a concentration-dependent manner (Figure 1A and Table 1). However, when the control experiments were performed in exactly the same manner but in the absence of cells (cell culture medium only, Figure 1B), the ROS formation kinetics were superimposable on the cell data (Figure 1A), suggesting that the DCFH_2_-DA → DCF conversion was mainly mediated by rotenone and not mitochondria. Moreover, cell culture medium alone was capable of deacetylating DCFH_2_-DA and oxidizing DCFH_2_, which is addressed elsewhere [31].

### 3.2. DCF Formation in Cultured Hepatocytes

Cells are normally grown under standard culture conditions in cell culture medium. The cell culture medium is a complex mixture of growth factors and nutrients that is designed to keep cells in an optimal condition over a longer period [80]. The nutritional requirements differ per cell type and function [80]. In contrast to cell culture medium, HBSS is commonly used to keep cells viable for a shorter period of time. HBSS contains inorganic salts that are supplemented with glucose and is buffered with phosphate to maintain a physiological pH and osmotic pressure [81,82]. PBS is an isotonic solution that can be used to wash cells and also has the ability to maintain osmolarity and pH [82]. Longer incubation with HBSS or PBS is not recommended because HBSS and PBS lack essential growth factors [81] and other vital constituents, which may perturb cell metabolism during experiments. 

All of the abovementioned solvents could theoretically be used as an assay medium in fluorescence experiments. However, since the nutritional composition differs between solvents, DCF formation most likely is a function of solvent composition. In order to design an assay system that minimizes undesired influences on DCF formation in cells, basal oxidant formation in hepatocytes that were incubated with cell culture medium (non-supplemented and phenol red-lacking WE medium containing 25 mM HEPES), HBSS, or PBS was measured. WST-1 conversion was used as measure for mitochondrial activity and cell death was assessed using a Hoechst assay.

In general, basal oxidant formation increased over time and was highest when the cells were incubated with PBS (Figure 2A–C). These results are in line with the lack of nutrients in PBS and are moreover supported by the fact that the extent of cell death was highest in all cell lines (Figure 2G–I). AML-12 cells had higher mitochondrial activity when they were incubated with PBS (Figure 2D), suggesting that ROS are excessively produced in the remaining cells. DCF fluorescence was similar in cells that were incubated with HBSS and cell culture medium (Figure 2A–C). However, HepG2 cells showed mitochondrial hyperreactivity after HBSS incubation (Figure 2F). Presumably, this higher mitochondrial activity is an early sign of mitochondrial stress that is caused by metabolic suppression, particularly in rapidly dividing cells such as HepG2 cells. These results echo previous results where HepG2 cells exhibited significantly higher DCF formation compared to HepaRG cells in the absence of ETC inhibition, which was ascribed to a higher metabolic rate of HepG2 cells [31]. DCF formation in kinetics experiments that were performed with cell culture medium yielded tapered curves for AML-12 (Figure 2A) and HepaRG cells (Figure 2B). HEPES, which is contained in the cell culture medium, was previously shown to deter DCF formation in cell-free experiments from 60 min incubation onward when compared to unbuffered medium [31]. A possible explanation for the curve tapering is therefore that DCF formation in the culture medium was deterred by HEPES at the later time points. The fact that fluorescence from HepG2 cells in assay medium is linear over the 2-h time period could be attributed to the high metabolic rate of HepG2 cells and corollary mitochondrial ROS production (Appendix A) stymieing the probe oxidation-ameliorating effects of HEPES.

Taking all of the abovementioned findings into consideration, PBS is an unsuitable medium for the fluorogenic probe loading stage. The cell culture medium was deemed more suitable than HBSS for fluorescence experiments because (1) ROS production occurred at near-equal rate in cells that were incubated with HBSS and medium, (2) DCF formation reached an equilibrium during the incubation period in two of the three cell lines that were maintained in the cell culture medium, and (3) the cell culture medium facilitates cell conditions that minimize basal oxidant formation while more optimally sustaining cell metabolism prior to subsequent experimental procedures (e.g., ETC inhibition experiments). Although DCF auto-oxidation did occur to a certain extent in cell culture medium, these effects are easy to correct for by subtracting data from a cell-free plate from the experimental plate, thus excluding culture medium-induced DCF formation [31]. The cell culture medium was therefore used as the assay medium in all subsequent experiments that concerned cells.

### 3.3. The DCFH_2_-DA → DCF Conversion Is Mediated by Mitochondrial Complex Inhibitors in Cell-Free Assay Medium

In light of the rotenone-induced formation of DCF from DCFH_2_-DA (Section 3.1), other mitochondrial complex inhibitors were tested for their propensity to convert DCFH_2_-DA to its fluorophore form. These included antimycin A, a commonly used complex III inhibitor that blocks the reduction of cytochrome c [83] (Appendix A). Myxothiazol and piericidin A were used as alternatives to rotenone, which also block the two-step reduction of Q to QH_2_ at complex I (Appendix A), albeit by different mechanisms. Myxothiazol prevents QH_2_ from funneling electrons from complex I to complex III. Although the exact site of action is currently elusive, myxothiazol reportedly acts on complex I as an QH_2_ antagonist [54] while also inhibiting electron transfer from QH_2_ to cytochrome c at complex III [84]. Piericidin A is a quinone antagonist and prevents the reduction of Q to QH [84]. The association between these inhibitors and their respective ETC binding sites triggers electron leakage from the ETC [76,85,86] that can be detected with DCFH_2_-DA. 

Table 1 shows the influence of the different ETC inhibitors on the formation of DCF in the absence of hepatocytes. In general, DCF fluorescence was strongly suppressed at acidic pH, as has been reported previously [28,31,87]. At basic pH, DCF fluorescence was only mildly suppressed; an effect that was observed previously [31] despite the higher absorptivity of DCF at increasing pH [31]. 

The ETC inhibitors typically did not cause DCF formation at concentrations of ≤ 10 μM (Appendix A) with the exception of piericidin A, where DCF formation occurred at > 1.0 μM concentration in unbuffered-, neutral buffered-, and alkaline-buffered WE medium (Appendix A). The most optimal incubation conditions were in neutral-buffered medium for all inhibitors, with the effect size of the probe conversion in the order of piericidin A > rotenone > myxothiazol ≈ antimycin A when corrected for medium-induced DCFH_2_ oxidation and inhibitor concentration. Rotenone, antimycin A, myxothiazol, and piericidin A could be used up to a concentration of 10 μM, 30 μM, 10 μM, and 1 μM, respectively, without inducing notable DCFH_2_ oxidation during 2-h incubation at 37 °C. The workable concentration ranges might be problematic given that higher inhibitor concentrations are exacted for some experiments (Appendix A).

### 3.4. Optimization of the DCFH_2_-DA Assay System

The initial goal of this study was to use ETC inhibitors to induce ROS production in three hepatocyte cell lines. However, rotenone, antimycin A, myxothiazol, and piericidin A all influenced DCF formation to some extent under different pH conditions in cell-free experiments (Section 3.1 and Section 3.2, Appendix A). Moreover, the ETC inhibitor solvents reduced the extent of DCF formation (Table 2, Appendix A, and [27,88]) and exerted solvent-, solvent concentration-, and cell type-dependent cytotoxicity (Appendix A). To enable correct data interpretation, DCFH_2_-DA should be added to cells separately from the ETC inhibitors. Also, a solvent should be used for the dissolution of ETC inhibitors (and other compounds that cells are exposed to in redox assays) that does not induce cell death at the added solvent concentration while reaching the desired inhibitor concentration in the test system. Solvent-induced execution of apoptotic pathways can in itself be a source of ROS [89] and hence offset the fluorescence-reducing effects of solvents on DCF readouts. These factors add layers of complexity to the experimental setup and could further thwart data interpretation.

During the initial experiments, the pH of the assay medium rose to 9.0 during the 2-h read in the plate reader (data not shown). Maintaining the pH at 7.4 is necessary since a pH of 9.0 is not compatible with cells and leads to deacetylation of DCFH_2_-DA, thereby accelerating DCF formation (Table 2, Appendix A, and [31]). To achieve a constant pH, a buffer can be added to the assay medium. The effect of 5–25 mM HEPES or TRIS buffer on DCF formation was therefore tested. These buffers did not influence ∆flu (Table 2, Appendix A). Accordingly, it was decided to buffer the assay medium with 25 mM HEPES.

The type of medium that is used during the assay also influences DCF formation [31,90,91]. Table 2 and Appendix A show that all of the tested culture media increased ∆flu to a certain extent over 2 h, although the amount of DCF formation differed between the different types of media. Especially DMEM, which is high in oxidant-generating compounds [31], significantly increased DCF formation over time in cell-free experiments. Therefore, DMEM is ill-suited for fluorescence experiments. Time-dependent DCF formation in RPMI and WE medium in cell-free experiments was comparable. As previous experiments were performed in WE-medium [31], allowing comparability of the results, and because WE-medium was originally designed for hepatocyte-like cell lines, WE-medium was used in the present study. To account for WE-medium-induced DCF-formation, the measured fluorescence in the cell experiments should be corrected with cell-free controls using the assay solution only [31,90]. However, proper data correction is not always described or performed [92,93,94]. Not performing cell-free controls can lead to erroneous conclusions on the magnitude and cellular localization of ROS formation. 

Serum is added to the culture media to facilitate cell growth and support cell metabolism. Metabolic alterations following serum deprivation only occurred after 24 h in human cancer prostate cell lines [95]. Immortalized hepatocytes (6/27 cell line) were shown to die two days after serum deprivation [96]. Serum is therefore not necessary per se in an assay medium to keep the cells viable for the duration of a 2-h kinetic read. Table 2 and Appendix A demonstrate that MilliQ-dissolved FCS, heat-inactivated FCS, and BSA increase DCF formation. These results underpin the findings of others that serum increases ∆flu when it is added to DMEM [91] or HBSS [97] and furnish an additional reason to use assay medium instead of fully-supplemented culture media.

Reiniers et al. [31,68] previously addressed an optimized protocol for measuring DCF formation in hepatocytes, albeit in the absence of ETC inhibitors. In line with previous results, we found that experiments should preferably be performed in a serum-free buffered medium (preferably not DMEM) so as to exclude changes in DCF formation by serum or pH. After performing the experiments, DCF data of a cell-free plate should be subtracted from the assay plate to correct for medium-induced DCF formation. The authors further suggested to use a DCFH_2_-DA concentration of ≥ 60 µM because DCF formation is concentration-dependent at lower concentrations. For future experiments, we therefore suggest using a final DCFH_2_-DA-concentration of 60–100 µM. However, the lower concentration that was used here does not detract from the central message, inasmuch this study aimed to demonstrate various protocol-related influences on the DCFH_2_-DA → DCF conversion in the context of ETC-inhibitors rather than to quantify general oxidative stress in hepatocyte cell lines. 

### 3.5. DCF Fluorescence and ETC Inhibition in Hepatocyte Cell Lines

After optimization of the DCFH_2_-DA assay system, AML-12, HepG2, and HepaRG cells were cultured in the presence of ETC inhibitors to induce mitochondrial ROS production and subsequently incubated with DCFH_2_-DA to analyze general oxidative stress. The DCF fluorescence was normalized to DNA content to account for ETC inhibitor toxicity-induced cell fallout. Readers should further note that the following experiments were designed to measure ROS in cells that were already oxidatively stressed, and not cells that experienced the onset of oxidative stress. For the latter experimental design, the same principles apply as delineated in this paper, with the main difference being that the cells should be loaded with fluorogenic probe and washed prior to exposure to ETC inhibitors. 

#### 3.5.1. Rotenone

In all three cell lines, 24 h of incubation with the highest rotenone concentration (100 μM) increased ∆flu (fluorescence at t = 120 min—t = 0 min) compared to the 0 μM rotenone group (Figure 3D–F). In AML-12 cells, incubation with lower rotenone concentrations (0–25 μM) for 6 and 24 h did not influence ∆flu significantly (Figure 3A,D). In HepG2 cells, incubation with rotenone for 6 h gave a lower ∆flu for all the rotenone concentrations (0–100 μM, Figure 3B). The same was observed at 25 μM and 2.5 μM rotenone after 24 h of incubation with HepG2 cells (Figure 3E). In HepaRG cells, a similar pattern was observed, where 2.5–100 μM of rotenone decreased ∆flu, compared to solvent control after 6 h of incubation (Figure 3C), and 0.5–25 μM rotenone decreased fluorescence compared to 0 μM rotenone after 24 h of incubation (Figure 3F). 

There are two hypotheses that could explain the fact that only incubation with 100 μM rotenone for 24 h increased ∆flu compared to the solvent control group in all cell lines, and that lower concentrations or a shorter incubation period (i.e., 6 h) decreased ∆flu in HepG2 and HepaRG cells and yielded no difference in AML-12 cells:

(I).At the lower rotenone concentrations, the cells are able to withstand the formation of mitochondrial ROS by upregulating their antioxidant enzyme systems (e.g., superoxide dismutase and catalase), but at higher concentrations (i.e., 100 μM rotenone) and longer incubation periods the antioxidant systems can no longer compensate [43], resulting in an increase in ∆flu. An increase in the antioxidative capacity not only effectively protects the cells from the formed ROS but might also induce a shift in the cellular redox state to a more reduced state, which might explain the drop in DCF formation at lower concentrations and shorter incubation periods with ETC inhibitors (Figure 3).(II).At lower rotenone concentrations, the cells stop using the ETC as their main energy source and meet their ATP demand by switching to glycolysis. Since most cancer cells have a strong predisposition for aerobic glycolysis for ATP production, this theory is supported by the fact that a decrease in fluorescence was measured in the two cancer-derived cell lines (HepG2 and HepaRG) and not in non-transformed AML-12 cells [98]. The increase in DCF formation at 100 μM rotenone in HepG2 and HepaRG cells can be explained by the fact that glycolysis alone is not sufficient to meet the cellular energy demand and that the cancer cells are partially respiring through the ETC (i.e., Warburg metabolism). At this rotenone concentration, the rest capacity of the ETC is blocked, resulting in ROS formation, and thus an increase in ∆flu.

#### 3.5.2. Antimycin A

A similar decreasing effect was observed when HepG2 and HepaRG cells were incubated with antimycin A (Figure 4). Antimycin A, a complex III inhibitor, is known to cause ROS production in isolated mitochondria [99,100]. However, in HepaRG cells, incubation with antimycin A for 6 or 24 h gave lower ∆flu for all antimycin A concentrations (0–30 μM) (Figure 4C,F), whereas in AML-12 and HepG2 cells antimycin A did not affect DCF formation (Figure 4A,B,D,E). Therefore, our results indicate that antimycin A did not induce ROS production in cultured hepatocytes. Nevertheless, we found that antimycin A is lethal when the cells were incubated for 48 h (data not shown), suggesting that antimycin A effectively blocks the ETC but causes cell death independently of mitochondrial ROS formation. A lack of ATP production by this blockage could, for example, cause cellular energy depletion, leading to necrosis [101].

#### 3.5.3. Myxothiazol and Piericidin A

Figure 5C shows that incubation with myxothiazol reduced DCF formation at all concentrations compared to 0 µM when HepaRG cells were incubated for 24 h (Figure 5B). No differences for incubation with myxothiazol were found in the 24-h incubated HepG2 and AML-12 cells (Figure 5A,B) Incubation with piericidin A for 24 h decreased ∆flu for all of the concentrations and in all the cell types compared to the 0 μM piericidin A group (Figure 5D–F). 

#### 3.5.4. Rotenone in Combination with Antimycin A

The combination of rotenone and antimycin A was added to the three tested cell lines to investigate whether simultaneously blocking complex I and III could provoke ROS production. The cells were incubated for 24 h with different concentrations of antimycin A, combined with 100 μM rotenone (i.e., the concentration that consistently increased DCF formation after 24 h of incubation). An increase in ∆flu was seen in AML-12 cells for the combination of 100 μM rotenone with 1.0–30 μM of antimycin A (Figure 6A). A similar increase was seen in HepG2 cells for 1.0 and 0.1 μM antimycin A combined with 100 μM rotenone (Figure 6B), and in HepaRG cells for all combinations (0.1–30 μM) of antimycin A with 100 μM rotenone (Figure 6C).

In all three tested cell types, incubation with rotenone (0–100 μM) or antimycin A (0–30 μM) for 48 h resulted in complete detachment of the monolayer (data not shown), indicating severe ETC inhibitor toxicity. Consequently, the 2-h measurement of DCF fluorescence could not be performed for this incubation period. 

In contrast to induction with antimycin A alone, the combination of rotenone (100 μM) and antimycin A (0.1–30 μM) increased the fluorescence in a linear fashion in all the cell lines (Figure 6). Previous reports have shown that the combination of rotenone and antimycin A increased H_2_O_2_ production due to reversed electron transport when isolated mitochondria were forced to respire exclusively on succinate (i.e., respiring through complex II) [51,102]. In addition, a study by Votyakova and Reynolds [53] demonstrated that the addition of antimycin A to rotenone led to an increase of H_2_O_2_ production when the ETC was fed with glutamate (a complex I substrate). However the H_2_O_2_ production rate returned to normal after 1–2 min [53]. It is, however, difficult to translate those effects from isolated mitochondria to cultured cells. The results that are shown in Figure 6 might be explained by two modes of ROS production at complex I, namely production at the FMN site of complex I following blockage with rotenone, and reversed electron transport due to respiration through complex II after the blockage of complex III with antimycin A.

## 4. Concluding Remarks

DCFH_2_-DA is a frequently used probe to analyze oxidative stress in vitro [27,31,91,103,104,105,106,107,108,109,110,111,112]. As DCFH_2_-DA to DCF conversion is catalyzed by a multitude of experimental variables such as pH and the type of medium that is used, an optimized protocol for the practical applicability of DCFH_2_-DA on hepatocyte cell lines was published previously [31,68]. In addition to that previous study, this study aimed to demonstrate the effect of various protocol-related factors on the DCFH_2_-DA → DCF reaction when oxidative stress is induced using ETC inhibitors. 

The most important finding was that all of the ETC inhibitors dose-dependently induced DCF formation in cell-free experiments. For this reason, induction of oxidative stress by ETC inhibitors should always be followed by incubation with DCFH_2_-DA, and both substances should not be combined simultaneously in a single solution. Moreover, in line with the previous study it was found that DCF formation is influenced by pH. Also, we concluded that serum-free medium should be used when performing fluorescence experiments since serum is otiose in short-term experiments and serum increases DCF fluorescence.

After the protocol was optimized, DCF fluorescence was measured after ETC inhibition. It was shown that rotenone only in the highest dosage that was used for 24 h increased ∆flu compared to the solvent control group in all of the cell lines. The fact that lower concentrations or a shorter incubation period (i.e., 6 h) decreased ∆flu in HepG2 and HepaRG cells and yielded no difference in AML-12 cells was ascribed to an upregulation of antioxidant enzymes and a switch to aerobic glycolysis, which was presumed to be most prominent in cancer-derived cell lines such as HepG2 and HepaRG. Consistent with incubation with rotenone at lower concentrations and/or for a shorter period, incubation with antimycin A, myxothiazol, and piericidin A decreased DCF formation in all the cell types. Strikingly, but in conformity with previous studies [51,53,102], the combination of antimycin A in various concentrations together with 100 μM rotenone increased DCF formation. 

The question remains whether DCFH_2_-DA is the right probe to measure ROS formation in mitochondria, which occurs in the inner mitochondrial membrane [113]. The most important drawback is the cytosolic localization of DCFH_2_. When (hydrophobic) DCFH_2_-DA enters the cell, it is deacetylated by esterases and forms the (hydrophilic) dye DCFH_2_ that is retained in the cytosol. Some ROS have a very short lifetime (e.g., •OH), and when mitochondrial production of these species is provoked with ETC inhibitors, those ROS probably react with biomolecules or antioxidant systems before they can reach the cytosol and convert DCFH_2_ to DCF. On the other hand, extraliposomal DCFH_2_ has been reported to react with ROS that is produced photochemically inside liposomal biomembranes [114] and Lee et al. [115] demonstrated DCF formation in illuminated cells that had been photosensitized with a mitochondria-specific iridium (III)-based photosensitizer. Accordingly, intracellular DCFH_2_ oxidation by mitochondria-derived ROS may very well occur. Nonetheless, mitochondrially-targeted probes such as MitoSOX might improve mitochondrial ROS detection. MitoSOX is derived from the fluorogenic probe dihydroethidium and, since MitoSOX itself is positively charged, it specifically accumulates in the mitochondrial matrix owing to the local negative membrane potential [116]. However, the use of MitoSOX also comes with limitations. Intracellular oxidants and enzymes such as cytochrome c and xanthine oxidase can, for example, oxidize MitoSOX before it is able to react with ROS and thereby reduce the efficacy of MitoSOX [117].

In addition, the specificity of DCFH_2_-DA towards certain ROS is controversial since the probe hardly reacts with superoxide, H_2_O_2_, and ONOOH/ONOO^−^ [21,22], but is easily oxidized by singlet oxygen [114] and tertiary species such as CO_3_^•−^, •NO_2_, •OH, and HOCl [21,22]. Just like MitoSOX, DCFH_2_-DA can be oxidized to DCF by cytochrome c, which is released from mitochondria during apoptosis [30,109]. Those influences, in combination with the fact that DCFH_2_-DA is not capable of measuring all of the formed ROS, makes it harder to make a statement about the amount of ROS and type of ROS that were formed in the described ETC inhibition experiments. Therefore, a change in ∆flu as a result of DCF formation should be interpreted as an indication of general oxidative stress rather than as a quantitative measure for ROS formation. 

Finally, this study did not investigate the potential difference in DCFH_2_-DA conversion dynamics in relation to the extracellular environment (medium composition and pH) versus the intracellular milieu (cytosolic constituents and pH) and in the context of cell physiology. These remain relevant issues in oxidative stress comparative analyses, especially if the conversion dynamics are differentially affected in the distinct spaces. The following hypothetical example illustrates the complexity of the particular issues and their (compounding) practical implications. There are two hepatocyte cell lines—primary rat hepatocytes (quiescent) and HepG2 cells (highly proliferative)—that were compared in terms of general oxidative stress following an exogenous stimulus (such as a ROS-generating drug intervention [118] or treatment modality [114]) in the framework of liver cancer treatment. During the DCFH_2_-DA incubation phase, the primary hepatocytes were incubated in Dulbecco’s modified Eagle medium (DMEM) [119] while the HepG2 cells were incubated in RPMI 1640 medium [120]. DCFH_2_-DA is converted to DCF at a higher rate in DMEM compared to RPMI 1640 medium [31,68], so the primary hepatocytes were exposed to more DCF and less DCFH_2_-DA than the HepG2 cells. If we assume that primary hepatocytes resemble differentiated HepaRG cells more so than cancer cells, then we can project that primary hepatocytes take up DCF more profoundly than HepG2 cells but export DCF at a near-equal rate [31,68]. Consequently, when DCF fluorescence is measured using a plate reader with bottom voxel configuration or by flow cytometry, the primary hepatocytes will reflect an overestimated ROS production due to extracellular DCF formation and cellular import. In a comparative analysis, the elevated DCF presence in primary hepatocytes may be offset by the fact that cancer cells such as HepG2 cells generally contain higher levels of esterases [121], whose catalytic activity is in part dictated by the molecular properties such as chirality [122]. Combined with the more profound ROS production in cells with faster metabolism and higher proliferation rate, the HepG2 cells will facilitate DCF formation due to intracellular probe conversion and ROS-mediated oxidation. However, these processes may in turn be offset by the fact that cancer cells generally possess greater levels of antioxidant enzymes [123] that may ameliorate the DCFH**_2_** → DCF conversion rate. Accordingly, in this example the readout parameter for general oxidative stress (i.e., DCF fluorescence) is affected by factors other than ROS (non-ROS probe conversion and trafficking) in primary hepatocytes and enzyme profiles in HepG2 cells, some of which are difficult to correct for in the experimental design. One of the reviewers of this article correctly pointed out that the present study did not investigate how the composition and pH of the medium affects the behavior of intracellular DCFH**_2_**-DA, implying that a simple washing step could solve some of the issues that were pinpointed through our experiments. Nevertheless, DCF formation in the medium and cellular uptake of the formed DCF cannot be solved by a simple washing step and is therefore also difficult to correct for. 

## 5. Conclusions

When inducing mitochondrial redox stress with ETC inhibitors, the experimental conditions (medium, pH, use of FCS, type of ETC inhibitor) have an effect on the outcome (i.e., ROS production and the corollary state of oxidative stress). Accordingly, to correctly interpret the data, the experimental design must account for these phenomena. This need is underscored by the finding that the ETC inhibitors themselves are capable of converting the non-fluorescent DCFH_2_-DA to the oxidized, fluorescent DCF. Furthermore, every cell line under investigation is idiosyncratic in terms of redox states and probe processing, and not all phenomena can be corrected for via experimental design. 

## Figures and Tables

**Figure 1 antioxidants-11-01424-f001:**
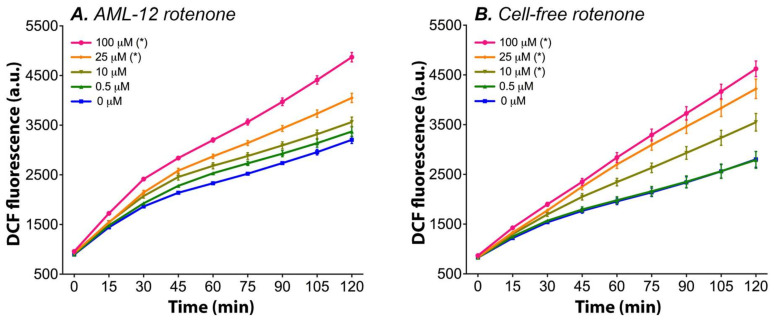
Rotenone-induced DCFH_2_-DA → DCF conversion kinetics in AML-12 hepatocytes (**A**) and in unbuffered assay medium (**B**). AML-12 hepatocytes received unbuffered assay medium (see Section 2.3 for details) containing 50 μM DCFH_2_-DA and either 0.5–100 μM rotenone or solvent control (0.1% (*v*/*v*) DMSO). DCF fluorescence was recorded for 2 h using a microplate reader that was equilibrated at 37 °C. The assays in medium are an exact duplicate of the experiments with hepatocytes, only in the absence of cells. The data are plotted as mean ± SEM of n = 4 experiments per rotenone concentration. (*) = *p* ≤ 0.05 relative to 0 µM rotenone (control) one-way ANOVA, Dunnett’s post hoc test. Abbreviations: DCF, 2′,7′-dichlorofluorescein; a.u., arbitrary units.

**Figure 2 antioxidants-11-01424-f002:**
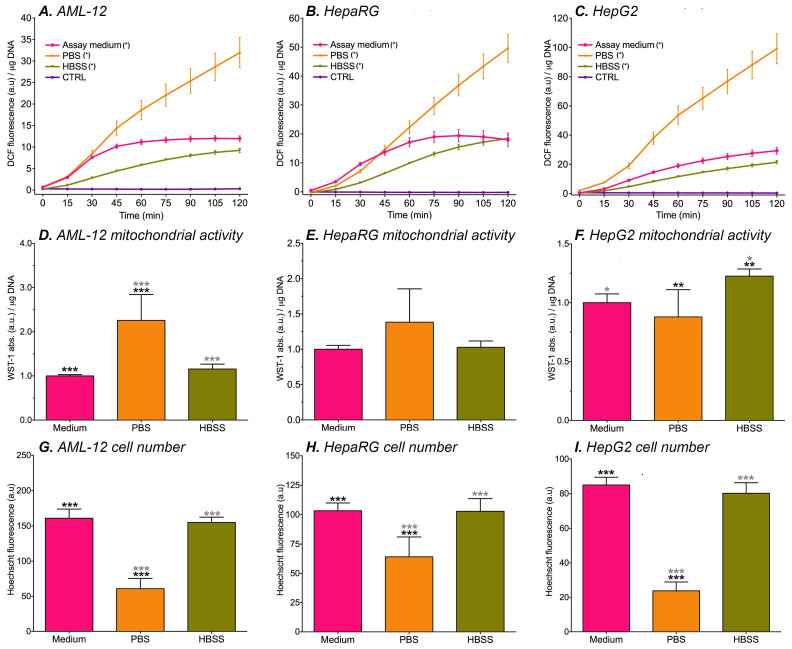
The influence of prolonged physiological buffer exposure on DCF fluorescence (indicating basal oxidant formation, **A**–**C**), WST-1 absorption (indicating mitochondrial activity, **D**–**F**), and Hoechst fluorescence (indicating cell number, **G**–**I**). The data are presented as mean ± SD (n = 6 per group). (*) = ∆flu (fluorescence at t = 120 − t = 0) significantly different vs. 0 μM (**A**–**C**, *p* < 0.05 one-way ANOVA, Dunnett’s post hoc test). * = *p* < 0.05, ** = *p* < 0.01, *** = *p* < 0.001 (**D**–**I**, one-way ANOVA, Tukey’s post hoc test). The assay medium was composed of non-supplemented and phenol red-lacking WE medium containing 25 mM HEPES. Abbreviations: DCF = 2′,7′-dichlorofluorescein; a.u. = arbitrary units; WST-1 = water soluble tetrazolium-1.

**Figure 3 antioxidants-11-01424-f003:**
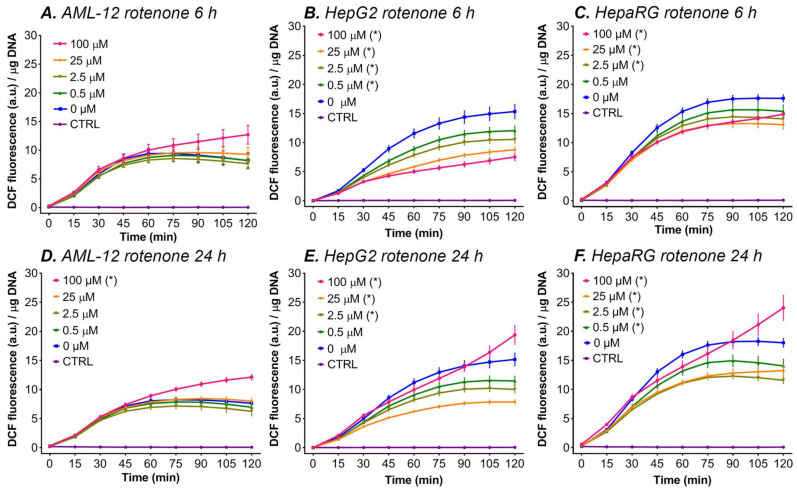
DCF fluorescence per μg DNA during a 2-h kinetic read after rotenone incubation for 6 or 24 h in AML-12, HepaRG, and HepG2 cells. The data are presented as mean ± SEM (n = 8 per group) for AML-12 and HepG2 cells and as the mean ± SEM (n ≥ 3/group) to increase graph legibility for HepaRG cells. (*) = ∆flu (fluorescence at t = 120 − t = 0) is significantly different vs. 0 μM (*p* < 0.05 one-way ANOVA, Dunnett’s post hoc test). The control (CTRL) cells were not incubated with DCFH_2_-DA. Abbreviations: DCF = 2′,7′-dichlorofluorescein; a.u. = arbitrary units.

**Figure 4 antioxidants-11-01424-f004:**
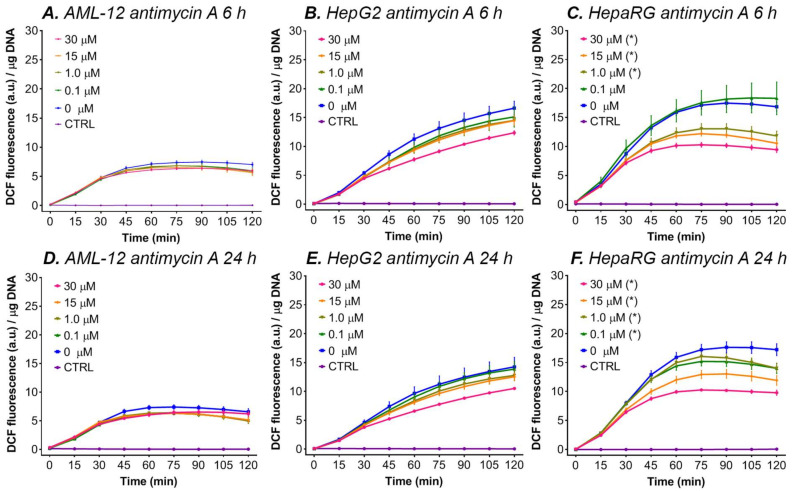
DCF fluorescence per μg DNA during a 2-h kinetic read after antimycin A incubation for 6 or 24 h in AML-12, HepaRG, and HepG2 cells. The data are presented as mean ± SEM (n = 6/group) for AML-12 cells, mean ± SEM (n ≥ 4/group) for HepaRG cells, and mean ± SEM (n = 4/group) for HepG2 cells to increase graph legibility. (*) = ∆flu (flu at t = 120 − t = 0) is significantly different vs. 0 μM (*p* < 0.05 one-way ANOVA, Dunnett’s post hoc test). The control (CTRL) cells were not incubated with DCFH_2_-DA. Abbreviations: DCF = 2′,7′-dichlorofluorescein; a.u. = arbitrary units.

**Figure 5 antioxidants-11-01424-f005:**
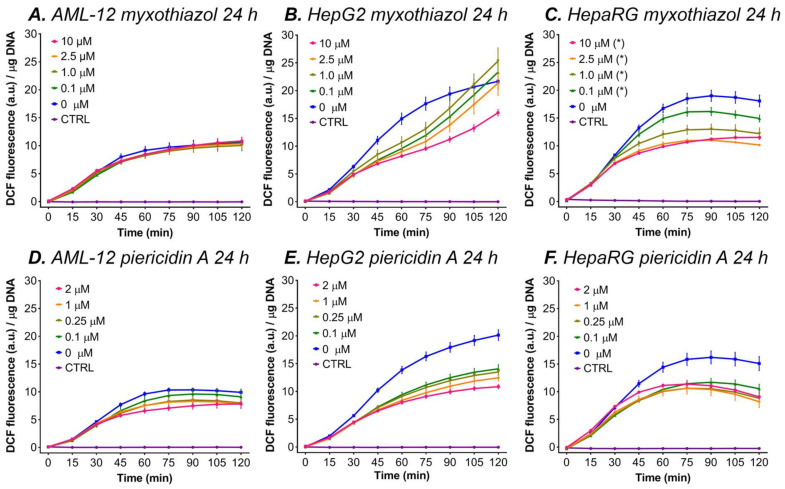
DCF fluorescence per μg DNA during a 2-h kinetic read after myxothiazol incubation for 24 h in AML-12, HepaRG, and HepG2 cells. The data are presented as mean ± SEM (n = 4/group) to increase graph legibility. (*) = ∆flu (flu at t = 120 − t = 0) is significantly different vs. 0 μM (*p* < 0.05 one-way ANOVA, Dunnett’s post hoc test). The control (CTRL) cells were not incubated with DCFH_2_-DA. Abbreviations: DCF = 2′,7′-dichlorofluorescein; a.u. = arbitrary units.

**Figure 6 antioxidants-11-01424-f006:**
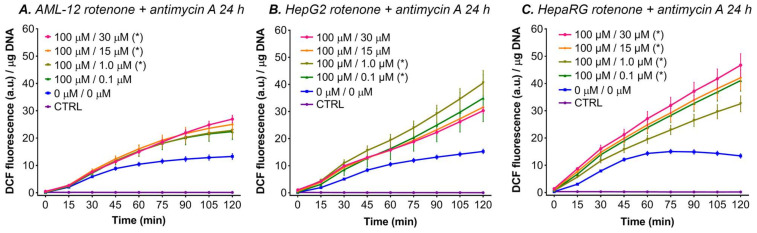
DCF fluorescence per μg DNA after incubation with rotenone and antimycin A for 24 h in AML-12, HepG2, and HepaRG cells. The data are presented as mean ± SEM (n = 4/group). (*) = ∆flu (flu at t = 120 − t = 0) significantly different vs. 0 μM (*p* < 0.05 one-way ANOVA, Dunnett’s post hoc test). The control (CTRL) cells were not incubated with DCFH_2_-DA. Abbreviations: DCF = 2′,7′-dichlorofluorescein; a.u. = arbitrary units.

**Table 1 antioxidants-11-01424-t001:** Influence of ETC inhibitors on the formation of DCF from DCFH_2_-DA under cell-free conditions. The values represent the mean ± SD change in DCF fluorescence (Δflu) in arbitrary units that were measured at the same spectrofluorometric settings (n = 8 per condition).

	Unbuffered	pH = 6.0	pH = 7.4	pH = 9.0
**Rotenone**				
0 μM	2395 ± 534	514 ± 45	2407 ± 332	1717 ± 306
0.5 μM	2363 ± 532	509 ± 49	2361 ± 346	1710 ± 319
10 μM	**3136** ± **551 ^#^**	**627** ± **62 ^#^**	2809 ± 384	**2288** ± **361 ^#^**
25 μM	**3846** ± **604 ^#^**	**621** ± **45 ^#^**	**3238** ± **330 ^#^**	**2713** ± **370 ^#^**
100 μM	**3996** ± **365 ^#^**	**658** ± **40 ^#^**	**4088** ± **333 ^#^**	**3294** ± **352 ^#^**
**Antimycin A**				
0 μM	2938 ± 331	727 ± 17	2340 ± 215	1520 ± 239
0.1 μM	2736 ± 388	677 ± 55	2314 ± 271	1453 ± 256
1.0 μM	2677 ± 442	712 ± 68	2351 ± 355	1369 ± 227
15 μM	**2277** ± **285 ^#^**	**985** ± **97 ^#^**	2299 ± 213	**1218** ± **113** **^#^**
30 μM	**2342** ± **284 ^#^**	**1211** ± **87 ^#^**	2335 ± 225	**1263** ± **89** **^#^**
**Myxothiazol**				
0 μM	1962 ± 363	696 ± 201	1898 ± 140	1519 ± 232
0.1 μM	1901 ± 376	673 ± 203	1741 ± 141	1443 ± 232
1.0 μM	1928 ± 336	705 ± 207	1760 ± 170	1540 ± 280
2.5 μM	2158 ± 392	736 ± 218	1846 ± 211	1753 ± 342
10 μM	**2631** ± **484 ^#^**	698 ± 183	2030 ± 145	**2325** ± **391 ^#^**
**Piericidin A**				
0 μM	2072 ± 176	124 ± 8.7	2311 ± 269	1123 ± 298
0.1 μM	2065 ± 170	127 ± 8.7	2296 ± 276	1071 ± 330
0.5 μM	2000 ± 184	**137** ± **7.1 ^#^**	2295 ± 303	1122 ± 368
1.0 μM	2042 ± 174	**155** ± **9.2 ^#^**	2402 ± 326	1179 ± 343
2.0 μM	**2331** ± **146 ^#^**	**184** ± **9.6 ^#^**	**2702** ± **303 ^#^**	**1778** ± **375 ^#^**

DCF fluorescence was recorded for 2 h using a plate reader (Section 2.3) and calculated by (DCF fluorescence at t = 120 min − DCF fluorescence at t = 0 min). (^#^, values in bold) indicates *p* ≤ 0.05 versus solvent control (0 μM) as tested by one-way ANOVA with Dunnett’s post hoc correction. All the experiments were performed in assay medium (non-supplemented and phenol red-lacking WE medium) buffered with 25 mM HEPES and adjusted to the indicated pH or in unbuffered assay medium. The DCF fluorescence kinetics traces are presented in Appendix A.

**Table 2 antioxidants-11-01424-t002:** The influence of assay components on the formation of DCF from DCFH_2_-DA under cell-free conditions. The values represent the mean ± SD change in DCF fluorescence (Δflu) in arbitrary units that were measured at the same spectrofluorometric settings (n = 8–16 per group).

Solvents ^†^	Ethanol	DMSO	Methanol	
0.0%	2646 ± 381	3005 ± 310	2884 ± 185	
0.2%	2033 ± 372	**2466** ± **314** **^#^**	**2546** ± **183** **^#^**	
0.5%	1902 ± 369	**2275** ± **333** **^#^**	**2434** ± **198** **^#^**	
1.0%	**1807** ± **384** **^#^**	**2148** ± **295** **^#^**	**2345** ± **179** **^#^**	
2.0%	**1738** ± **390** **^#^**	**1979** ± **286** **^#^**	**2342** ± **168** **^#^**	
4.0%	**1848** ± **374** **^#^**	**2066** ± **232** **^#^**	**2299** ± **149** **^#^**	
**Serum** **Buffers** ** ^‡^ **	FCS	FCS-HI	BSA	
0.0%	10 ± 5	17 ± 4	−12 ± 5	
1.0%	**564** ± **57** **^#^**	**502** ± **65** **^#^**	**328** ± **27** **^#^**	
2.5%	**930** ± **101** **^#^**	**719** ± **147** **^#^**	**303** ± **36** **^#^**	
5.0%	**873** ± **144** **^#^**	**781** ± **185** **^#^**	**157** ± **25** **^#^**	
7.5%	**607** ± **105** **^#^**	**610** ± **186** **^#^**	**127** ± **18** **^#^**	
10.0%	**425** ± **61** **^#^**	**475** ± **155** **^#^**	**147** ± **14** **^#^**	
**Buffers** ** ^‡^ **	TRIS	HEPES	HEPES [25 mM]	
5 mM	10.3 ± 14.8	4.4 ± 14.3	pH = 6.0	−8.9 ± 8.1
10 mM	3.8 ± 13.8	1.5 ± 15.1	pH = 7.4	−14.4 ± 4.9
25 mM	1.6 ± 14.6	2.1 ± 14.4	pH = 9.0	117.0 ± 40.0 ^a^
**Culture medium**	WE	DMEM	RPMI	DMEM/F12
	**2340** ± **215 ^c^**	**4017** ± **471 ^b^**	**1592** ± **140 ^c^**	**1834** ± **150 ^b^**

DCF fluorescence was recorded for 2 h using a microplate reader (Section 2.3) and calculated by (DCF fluorescence at t = 120 min − DCF fluorescence at t = 0 min). All the solutions contained 25 mM HEPES and were set to pH = 7.4 unless otherwise indicated. ^#^ signifies *p* < 0.05 versus the control group (i.e., no solvent, FCS, or albumin added), ^a^ indicates *p* < 0.05 versus the other pH values, ^b^ indicates *p* < 0.05 versus the other culture media, and ^c^ indicates *p* < 0.05 versus WE and DMEM. All the intergroup and intragroup differences were tested with a one-way ANOVA with Dunnett’s or Tukey’s post hoc correction (Section 2.6). The experiments were performed in assay medium (^†^) or MilliQ water (^‡^). The DCF fluorescence kinetics traces are presented in Appendix A. Abbreviations: FCS, fetal calf serum; FCS-HI, heat-inactivated fetal calf serum; BSA, bovine serum albumin.

## Data Availability

The data are contained within the article and Appendix A.

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
