# Peer review of "Experimental Conditions That Influence the Utility of 2′7′-Dichlorodihydrofluorescein Diacetate (DCFH2-DA) as a Fluorogenic Biosensor for Mitochondrial Redox Status"

_antioxidants, 2022, doi:10.3390/antiox11081424_

Round 1

Reviewer 1 Report

Page 9 line 347 the word stimieing should be spelt stymieing.

References Pages19-23 All lines in the references 4-113 should be justified as references 1-4.

Supplementary Information Page 2 the structure of DCF has two pentavalent carbons. The double bonds should be shifted to show the correct structure as depicted in the attached file

Reviewer 2 Report

De Haan et al. submitted a manuscript to Antioxidants dealing with the fluorogenic biosensor DCFH2-DA broadly used to evaluate the level of ROS in cells. Herein DCFH2-DA was employed to determine the extent of ROS production in hepatocytes exposed to different ETC inhibitors.

The authors conducted a comprehensive array of experiments reproducing the conditions typically used in cell cultures to screen the effect of different variables on the fluorescence of the probe. Even if the subject of this paper can appear as trivial at first since DCFH2-DA has been used for decades to assay ROS in cellular contexts, the authors point out different situations where the experimental conditions may lead to erroneous data interpretation. They further provide a very useful roadmap to bypass the pitfalls associated to this sensor. They also suggest alternatives to the use of DCFH2-DA such as MitoSOX. I therefore highly recommend the publication of this paper to Antioxidants as a useful complement to the paper published in 2021 in the same journal (ref 31).

I noticed the following typos:

Lines 79, 82: dichlorodihydrofluorescein

Line 347: stymieing

The authors might want to quote a tutorial paper dealing with the assay of ROS with the same probe (Methods 2016, 109, 3-11)

Reviewer 3 Report

The paper “Experimental conditions that influence the utility of 2’7’-dichlorodihydrofluorescein diacetate (DCFH2-DA) as a fluorogenic biosensor for mitochondrial redox status”, proposed for publication in Antioxidants provide an interesting work on the effects of various factors on the ROS detection by means of DCFH2-DA-DCF reaction, with also valuable advices on the application of the procedure to the assay of oxidative stress.

The paper in general is well written, experiments are conducted properly and results in general seem convincing, and only minor points need attention, as detailed below. 

Minor remarks

Line 100-101, the sentence is unclear; please improve its meaning “..and rotenone-primed cell-free medium…”

In the Tables, it is suggested to give the SD as the most classical “± “ rather than inside squared brackets

Reviewer 4 Report

This manuscript “Experimental conditions that influence the utility of 2’7’-di-2 chlorodihydrofluorescein diacetate (DCFH2-DA) as a fluoro-3 genic biosensor for mitochondrial redox status” by Lianne R. de Haan et al. describes fluorogenic probe 2´,7´-dichlorodihydrofluorescein-diacetate (DCFH2-DA).

The fundamental question of how the composition and pH of the medium affect the behavior of intracellular DCFH2-DA is not addressed at all. Even if it does affect the probes in the medium, why not just wash the cells before measurement?

Round 2

Reviewer 1 Report

No further comments

Reviewer 2 Report

The paper is now acceptable for publication to antioxidants

Reviewer 3 Report

The paper “Experimental conditions that influence the utility of 2’7’-dichlorodihydrofluorescein diacetate (DCFH2-DA) as a fluorogenic biosensor for mitochondrial redox status”, has been duly revised according to the minor points requiring attention, and can be now accepted for publication in “antioxidants”

Reviewer 4 Report

Now the revised one is aceptable in antioxidants.